# Parasites Circulating in Wild Synanthropic Capybaras (*Hydrochoerus hydrochaeris*): A One Health Approach

**DOI:** 10.3390/pathogens10091152

**Published:** 2021-09-07

**Authors:** Manuel Uribe, Carlos Hermosilla, Arlex Rodríguez-Durán, Juan Vélez, Sara López-Osorio, Jenny J. Chaparro-Gutiérrez, Jesús A. Cortés-Vecino

**Affiliations:** 1Biomedical Research Center Seltersberg (BFS), Institute of Parasitology, Justus Liebig University Giessen, 35392 Gießen, Germany; mmanuel.uribe@udea.edu.co (M.U.); carlos.r.hermosilla@vetmed.uni-giessen.de (C.H.); juan.velez@vetmed.uni-giessen.de (J.V.); 2CIBAV Research Group, Veterinary Medicine School, Universidad de Antioquia, Medellín 050034, Colombia; sara.lopezo@udea.edu.co (S.L.-O.); jenny.chaparro@udea.edu.co (J.J.C.-G.); 3Grupo de Investigación Parasitología Veterinaria, Facultad de Medicina Veterinaria y de Zootecnia, Universidad Nacional de Colombia, Bogotá D.C. 111321, Colombia; arrodriguezdu@unal.edu.co

**Keywords:** capybara, *Echinocoleus hydrochoeri*, *Plagiorchis muris*, *Neobalantidium coli*, *Cryptosporidium*, zoonoses, rodents

## Abstract

Capybaras (*Hydrochoerus hydrochaeris*) are affected by a wide range of protozoan and metazoan-derived parasitic diseases. Among parasites of free-ranging capybaras are soil-, water-, food- and gastropod-borne parasitosis, today considered as opportunistic infections in semiaquatic ecosystems. The overlapping of the capybara’s natural ecological habitats with human and domestic animal activities has unfortunately increased in recent decades, thereby enhancing possible cross- or spillover events of zoonotic parasites. Due to this, three synanthropic wild capybara populations in the Orinoco Basin were studied for the occurrence of gastrointestinal parasite infections. A total of forty-six fecal samples were collected from free-ranging capybaras in close proximity to livestock farms. Macroscopical analyses, standard copromicroscopical techniques, coproELISA, PCR, and phylogenetic analysis revealed thirteen parasite taxa. In detail, the study indicates stages of five protozoans, four nematodes, one cestode, and three trematodes. Two zoonotic parasites were identified (i.e., *Plagorchis muris*, and *Neobalantidium coli*). The trematode *P. muris* represents the first report within South America. In addition, this report expands the geographical distribution range of echinocoelosis (*Echinocoleus hydrochoeri*). Overall, parasitological findings include two new host records (i.e., *P. muris*, and *Entamoeba*). The present findings collectively constitute baseline data for future monitoring of wildlife-derived anthropozoonotic parasites and call for future research on the health and the ecological impact of this largest semiaquatic rodent closely linked to humans, domestic and wild animals.

## 1. Introduction

The Rodentia order is the most numerous and diverse within the Mammalia class composing up to 42% of worldwide mammalian biodiversity with at least 2277 species reported so far [1]. Rodents are distributed all over the continents except for Antarctica [2]. Capybaras (*Hydrochoerus*, Rodentia: Caviidae) are New World semiaquatic and herbivorous hystricomorphs, which represent the world’s largest and heaviest species among extant rodents [3]. The genus *Hydrochoerus* includes two species, namely *H. hydrochaeris* and *H. isthmius*. The first one is conspicuously larger and broadly distributed across South America, and the second one is an endemic species of Colombia, Panama, and Venezuela [4]. The biology of this giant rodent is closely linked to water sources such as rivers, swamps, natural lakes, and manmade water cumulus for domestic animal usage [5].

Moreover, capybaras are synanthropic species distributed in riparian habitats with strong anthropogenic impact and representing an important bushmeat source for traditional communities [6]. Accordingly, the Orinoco River constitutes a huge tropical watershed with extensive wetlands, marshes, and lakes nourished by the fourth largest river in the world. Unfortunately, this diverse and unique ecosystem has suffered dramatic anthropogenic changes during the last decades because of legal and illegal agroindustrial crop development [7], the expansion of the agricultural frontier due to livestock, illegal mining, logging [8], hunting [6], and wildlife trafficking [9]. Thus, free-ranging capybaras distributed in the human-altered Orinoco Basin are constantly interacting with a plethora of endemic wildlife species, humans, and domestic animals such as horses (*Equus caballus*), cattle (*Bos taurus indicus*), chickens (*Gallus gallus domesticus*), dogs (*Canis familiaris*), and cats (*Felis catus*). Thus, this demands One Health concept-oriented attention on parasitic infections. In this sense, a particular interplay between humans, animals, and the environment may lead to the emergence and transmission of zoonotic diseases as is highlighted by the increased transmission of wildlife-origin pathogenic agents [10]. Consistently, it seems essential to monitor parasitic infections in synanthropic species such as capybaras within their natural habitats, to identify potential zoonotic parasites and their impact not only on human health but also on domestic animals and endemic wildlife populations.

Currently, both endogenous proto- and metazoan parasites have been described in capybaras (Table 1) but little attention has been paid to zoonotic infections, particularly soil-, water-, food- and gasteropod-borne parasites. Therefore, the present study aims to describe the gastrointestinal parasite fauna of wild synanthropic capybara populations in the Orinoco Basin.

## 2. Results

Overall, thirteen gastrointestinal parasite taxa were found comprising five protozoans of the phyla Apicomplexa, Amoebozoa, and Ciliophora, and eight metazoans of the phyla Nematoda and Platyhelminthes (Class: Cestoda and Trematoda), thereby covering a rather wide range of parasites when compared to previous studies (see Table 1).

### 2.1. Fecal Macroscopical Examination

The feces’ gross examination revealed three helminths (i.e., one nematode and two trematodes). Nematodes found in Cinaruco were identified as *Protozoophaga obesa* female pinworms based on taxonomic traits (see Figure 1). The collected specimens had straight caudal extremity and an evident thin tail uncoiled posterior end. The mean morphometric measurements (*n* = 6) obtained were total body length 3.21 mm, body width 227.69 µm, and a total tail length of 815.72 µm. The mouth opening had four oxyurid-characteristic lips and four corresponding amphids. A straight thick short cylindrical oesophagus (118.25 µm in length) was observed with a wide cavity and a poorly chitinized pyriformous oesophagus bulb (49.85 µm × 48.89 µm). A wide straight intestine in the anterior portion of the body, which ended slightly contorted posteriorly was noticed.

Furthermore, an elongated reddish trematode was found in La Maporita collected feces (Appendix A). Studies of wet-mounted specimen and dark-ground stereomicroscopy evidenced asymmetric rows of spines along the integument decreasing in size towards the posterior region. The subterminal position of the oral sucker presented two notorious bilateral papillae. The morphometric data obtained from body measurements were 9.34 mm of total body length and 1.64 mm in width. The oral sucker was slightly ellipsoidal (74.87 µm × 80.76 µm). A narrow excretory terminal pore was also evidenced. Therefore, the digenean was identified such as *Hippocrepis hippocrepis* as previously described elsewhere [29]. Additionally, a dehydrated digenean specimen was found inside a fecal pellet in Bocas del Arauca. Due to poor conservation the specimen was thereafter subjected to subsequent phylogenic analysis.

### 2.2. Microscopical, Coproantigen, and Molecular Parasite Identification

The copromicroscopical analysis (*n* = 46) revealed nine parasite taxa comprising three protozoans and nine metazoans. Unsporulated *Eimeria*-like oocysts were detected and bilayered oocyst walls. Both oocyst wall layers (OWL) were smooth, with the outer OWL slightly yellowish and the inner OWL darker. The above depicted traits of the identified oocysts corresponded well with *Eimeria trinidadensis* (Figure 2a). Likewise, *Echinocoleus hydrochoeri* brownish barrel-shape plugged eggs were identified (Figure 2b and Appendix A). In addition, *P. obesa*-eggs were also detected (Figure 2c). Cestode eggs of *Monoecocestus* were found in a low number (Table 2); the eggs showed typical embryphore with a pyriformous apparatus containing a developed hexacanth embryo (oncosphere; Figure 2d). Furthermore, *Neobalantidium coli* cysts were identified (Figure 2e). In addition, two trematode species were found, i.e., *Hippocrepis hippocrepis* (Figure 2f and Appendix A), eggs with typical long and bilateral filaments in the poles of a capsulated miracidium (Figure 2g), as well as *Taxorchis schistocotyle* eggs (Figure 2h). Rhabditiform larvae of *Strongyloides* (Figure 2i,j) were also identified. Moreover, ascarid eggs were found (Figure 2k). Additionally, *Entamoeba* immature cysts were detected (Figure 2l).

Coproantigen ELISA showed a *Cryptosporidium* occurrence of 34.8% (16/46). Cinaruco had the highest *Cryptosporidium* occurrence (6/8), followed by La Maportia and Bocas del Arauca [(8/23) and (2/15), respectively]. In contrast, no *Giardia*-derived antigen was here detected. The complete list of parasitic stages, detection technique, occurrence, and geographic locale are summarized in Table 2. Moreover, though outside the scope of the current study, it is worth mentioning that a wide variety of commensal ciliated protozoans belonging to the Cycloposthiidae family were found (refer to Appendix A). Additionally, different shapes and colors of microplastic fibers in most analyzed samples were evidenced.

In order to identify the dehydrated digenean specimen found in capybara feces, the ~1300 bp-long fragments of 28S rDNA gene subjected to phylogenetic analysis showed that the analyzed specimen clustered within the lineage composed of the representative of *Plagiorchis muris* (Figure 3).

## 3. Discussion

Synanthropic populations of free-ranging capybaras are associated not only with cattle farming but also with human settlements. Additionally, this giant rodent is an important meat source for countryside populations [6]. It is estimated that approximately 61% of human diseases are zoonotic, and wildlife reservoirs are the source of most human emerging infectious diseases [40,41]. Consistently, capybaras have been reported as natural reservoirs of various zoonotic pathogens such as the tick-borne Brazilian spotted fever (BSF) [42], the liver fluke *Fasciola hepatica* [27], and the enteropathogen *Cryptosporidium parvum* [31]. This highlights the role of capybaras in the ecoepidemiology of infectious diseases. Thus, free-ranging capybaras should be considered as reservoir hosts for various zoonotic-relevant water-, food-, soil- and gastropod-borne parasites of public health concern.

Literature reporting on Colombian capybara parasitic infections is scarce and restricted to solely two four decades old reports, which describe microfilariae [11,12]. Thus, neither morphological data nor molecular descriptions have been reported for gastrointestinal parasites occurring in Colombian capybara populations. To the best of our knowledge, we present for the first time a wide gastrointestinal parasite study of free-ranging capybaras. Overall, the current study revealed thirteen parasite taxa, five protozoans (i.e., *Cryptosporidium*, *E. trinidadensis*, *Entamoeba*, *N. coli*, and *Cyclophosthium* sp.), four nematodes [i.e., *E. hydrochoerid* (Trichuridae), *P. obesa* (Oxyuridae), strongyloid-like (Strongyloididae), and Ascarididae], one cestode [*Monoecocestus* sp. (Anoplocephalidae)], and three digenean trematodes [i.e., *H. hippocrepis* (Notocotylidae), *P. muris* (Plagiorchiida), and *T. schistocotyle* (Cladorchiidae)]. 

Here we identify *P. muris* infecting capybaras for the first time. Within the Plagiorchiidae family, this is the only species capable of infecting humans and it has been reported across continents [43]. This neglected trematode infects mainly humans from Korea, Japan, and North America proving an effective transmission route from rodents to humans [44,45,46,47,48]. Often reported as a rodent-borne disease with public health concern in mainland southeast Asia [49], Iran [50] and the Netherlands [51], this trematode has never been reported in South America before. Despite its well characterized occurrence and zoonotic potential across Asian countries, detailed information on epidemiology, life cycle, and reports on human plagiorchiosis remain still scarce in Africa, the Americas, and Europe [51,52,53]. Thus, the present description expands the previously known geographic distribution range of this parasite and constitutes the first host record of capybaras. Meanwhile, *P. muris* has been reported to infect cats, dogs, and chickens [54,55,56]. Therefore, we recommend future activities on this euryxenous zoonotic trematode in South American domestic animals and humans for a better life cycle comprehension, potential obligate first gastropod intermediate host spectrum, second intermediate host, other final hosts, and possible public health impact of plagiorchiosis in rural populations.

Moreover, surveillance of zoonotic parasitic agents by local public health authorities should be recommended as capybaras are still frequently consumed [6]. Notably, bushmeat-related activities have been linked to parasitic emerging diseases outbreaks to human populations such as neobalantidiosis, amoebiasis, strongyloidiosis, and giardiasis [57]; it is worth mentioning that the present study has shown a general coinfection occurrence of 17.4% (8/46) for *H. hippocrepis*, *P. obesa,* and *Entamoeba* in capybaras. Since *E. histolytica* is a cosmopolitan extracellular enteric parasite causing amoebiasis with an average of 50 million cases and 55,000 to 100,000 human deaths each year globally [58], further monitoring of this zoonosis in capybara populations should be addressed. Interestingly, the water/food-borne parasite *N. coli* was detected with low occurrence in Bocas del Arauca and La Maporita. This enteropathogenic ciliated protozoa can be found throughout the world infecting mainly pigs, wild boars (*Sus scrofa*), rodents, equines, ruminants, nonhuman primates, and humans [59,60]. It is possible that *N. coli* cysts have a multidirectional interchange and transmission among humans, capybaras, and domestic animals. Additionally, a general occurrence of 41.3% (19/46) for *Strongyloides*-like larvae were reported. The strongyloidiosis is a parasitic disease caused by nematodes in the genus *Strongyloides* that remain largely neglected [61]. Thus, infected capybara herds may contribute to the dissemination of related soil-borne nematodes.

Overall, the highest parasite occurrence found in capybaras were recorded for *E. hydrochoeri* [(58.7% (27/46)] and the trematode *T. schistocotyle* [43.5% (20/46)]. The capillariid genus *Echinocoleus* has been reported in natural populations of capybaras in the northeast Argentinian Iberá Wetlands [22]. Despite the difficulty of studying this nematode group, the paucity of good morphological characteristics, and complex systematics [24], *E*. *hydrochoeri* is widely distributed in Brazil from the Pantanal, Mato Grosso do Sul, to Rio Grande do Sul where it has been identified in capybaras linked to cattle breeding areas [15,17]. Irrespectively, the present study constitutes the first report of this parasite in Colombia, thus expanding the distribution range of echinocoelosis to the Orinoco Basin. Capybara echinocoelosis may result in gastrointestinal disturbances due to enteric and intestinal location of pre- and adult stages [24]. Additionally, *T. schistocotyle* has been associated with multifocal necrotizing colitis in capybaras [62], and it is worth considering as a major concomitant lethal cause in heavily infected rodents, as well as fasciolosis [27].

Furthermore, *Cryptosporidium*-specific antigens (CSA) were found in studied areas, thus indicating a potentially wide occurrence along the Orinoco Basin. While anthropozoonotic *Cryptosporidium* species have already been described in nine rodent families [63,64,65,66,67,68], there is only one report of *C*. *parvum* in capybaras [31]. Interestingly, the highest occurrence of CSA was found in Cinaruco, furthest away from populated centers whose closest human settlement is located at 20.598 Km. In contrast, areas with greater anthropogenic effect show lower occurrence. Since the hypertransmissible IIaA15G2R1 *Cryptosporidium* subtype naturally infect Brazilian wild capybaras [31], and the oocysts could easily spread in aquatic/semiaquatic ecosystems [69], it is necessary to identify the species/genotype and the related zoonotic potential of *Cryptosporidium* detected in capybaras in this study. 

The present findings collectively reflect the parasitological status of wild synanthropic capybara populations in the Orinoco Basin, a well-known neotropical grassland region for extensive cattle production. Therefore, this region faces a special risk of parasites transmission among capybaras, wild/domestic animals, and humans. Based on these results, we encourage further parasite monitoring studies on wild capybara populations. For comparative reasons, parasitological surveys of the species *H. isthmius* should be considered in the near future. Further investigations are required to reveal the importance of described parasites not only for public health concern but also for neotropical wildlife conservation issues. Since detected trematodes (i.e., *P. muris, H. hippocrepis*, and *T. schistocotyle*) require either terrestrial/amphibious or aquatic obligate gastropod intermediate hosts, it would be appropriate to analyze in depth the gastropod fauna (e.g., snails and slugs) inhabiting flooded areas, rivers, natural lakes, and ponds shared by humans, domestic animals, and wildlife to prevent zooanthroponotic parasite spillovers.

## 4. Materials and Methods

### 4.1. Study Areas and Sample Collection

Three wild populations of capybaras were investigated in west-side lowlands of the Orinoco Basin (Figure 4). The geographic study areas have an average annual precipitation of 1477 mm, 90% relative humidity, 120 m above sea level, and a temperature range between 27.8 and 30.9 °C. In accordance with the Köppen–Geiger climate classification sampling areas were located in tropical savannah [70]. Individual fecal samples were collected during dry season from March to June 2020 in La Maporita (06°55′37.77″ N; 070°27′46.54″ W), Cinaruco (06°40′46.88″ N; 070°7′9.36″ W), and Bocas del Arauca (07°01′10.31″ N; 070°35′28.97″ W), Arauca, Colombia.

The noninvasive methodology allows the collection of feces after spontaneous defecation without disturbing the social dynamics and natural behavior of capybara herds as has been successfully performed for other wild mammals [71]. Free-ranging capybaras feces (*n* = 46) were randomly collected from manure pellet piles regardless of the gender and age (Figure 5). Fecal samples were immediately placed in 50 mL conical tubes (Sarstedt, Nümbrecht, Germany) containing 80% EtOH for fixation until further analyses. Helminth stages spontaneously shed within feces were gently rinsed in sterile 0.9% PSS and afterwards fixed in 96% EtOH for subsequent taxonomic and molecular identification. Sampling procedures were conducted in agreement with the international guidelines for the use of wild mammal species in research [72], the EU Directive 2010/63/EU, and the approvement of Ethics Committee for Animal Experimentation of the Universidad de Antioquia (AS Nº 131), Colombia.

### 4.2. Parasitological and Immunological Analysis

Fecal samples were processed following standardized parasitological techniques: combined sedimentation–flotation (SF), modified sodium acetate–acetic acid–formalin (SAF) [73], simple sedimentation (SS) [74], centrifugal flotation with zinc sulfate (CF), and fast carbol–fuchsin stained fecal smears (CFS) for *Cryptosporidium-*oocyst detection [75]. Copromicroscopical findings of parasitological stages, such as cysts, oocysts, sporocysts, eggs, and larvae were identified through corresponding morphology and morphometry analyses under an Olympus BX53 (Olympus Corporation, Tokyo, Japan) semimotorized light microscope equipped with an Olympus DP74 digital camera and the *cellSens* standard imaging software. Macroscopic parasites were identified using an Olympus SZX7 (Olympus Corporation, Tokyo, Japan) stereomicroscope system with Olympus DP27 digital camera and the above-described software. The parasite stages’ identification was based on general morphology, shape, size, and color, according to [24,29,33,62,76,77,78]. 

Furthermore, commercially available coproantigen ELISA for detection of *Cryptosporidium*- and *Giardia-*specific antigens (CSA and GSA) were performed by the membrane-bounds solid phase immunoassays ProSpecT *Cryptosporidium* Microplate Assay and *Giardia* Microplate Assay (Thermo Scientific, Waltham, MA, USA; Oxoid, Basingstoke, UK), following the manufacturer’s instructions. The interpretation of the results considered as negative all colorless reactions via visual reading and/or when OD values were < 0.05 after the blank of negative controls, indicating none or undetectable levels of GSA/CSA in analyzed samples. On the other hand, a variable intensity of yellow was considered as positive when OD values were > 0.05.

### 4.3. Molecular Analyses 

To characterize helminths found in feces a small sample of the midsection body tissue (≈10 mg) was dissected to extract gDNA using the DNeasy Blood and Tissue kit (Qiagen, Dusseldorf, Germany) following the manufacturer´s instructions. A ~1300 bp fragment of 28S rDNA gen was PCR-amplified using the primers: for-5′-aagcatatcactaagcgg-3′, and rev-5′-gctatcctgagggaaacttcg-3′, following the thermocycle profiles previously described [79]. The 28S rDNA gene was selected considering it is a more conserved gene compared to the mtCOI region among the Plagiorchiidae family [80]. All PCR products were bidirectionally sequenced by LGC Biosearch Technologies (Berlin, Germany). Representative 28S rDNA sequences of the recognized extant species were included in order to reveal the phylogenetic relationships. *Dicrocoelium dendriticum* [JQ081959; (Dicrocoeliidae)] was here used as the outgroup digenean species. SeqManPro 7.1.0 (DNASTAR Inc., Madison, WI, USA) was used to in silico edit, and finally assembled the sequence. The 28S rDNA alignment were conducted using the online version of MAFFTv. 7 (available at https://mafft.cbrc.jp/alignment/server/ accessed on 21 February 2020) [81]. A Neighbor-Joining algorithm analysis under 1000 bootstrap replicates was conducted in MEGAX software. Nucleotide sequence divergences were calculated using the Kimura2-parameter (K2P) model for multiple substitution distance correction and were in the units of the number of base substitutions per site [82]. The bootstrap consensus tree inferred from 1000 replicates was taken to represent the evolutionary history of the analyzed helminth [83]. Branches corresponding to partitions reproduced in less than 50% bootstrap replicates were collapsed.

## 5. Conclusions

Capybaras are effective indirect indicators of ecosystem health and therefore helpful to better understand potential parasites transmission routes to humans and domestic animal populations. Here we report the presence of thirteen parasite taxa in capybaras, some of them baring zoonotic potential. Thus, this rodent should be constantly monitored as public health concern issue. The noninvasive sampling methodology allows detection of large number of protozoan and metazoan parasites without altering wildlife populations. It is very important to broaden these epidemiological studies to other wildlife species inhabiting the Orinoco Basin and more frequent parasitological survey of this rodents within South America, since accurate identification of soil-, water-, food-, and gastropod-borne parasites will contribute to early detect and prevent parasitic spillovers events within the One Health concept.

## Figures and Tables

**Figure 1 pathogens-10-01152-f001:**
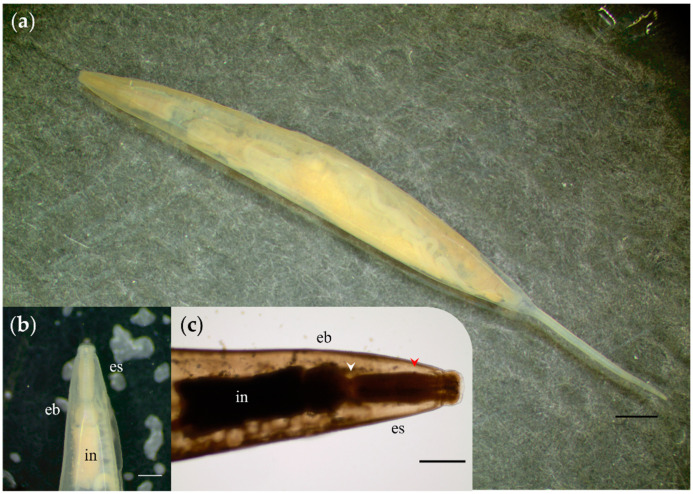
Female adult specimen of *Protozoophaga obesa*. (**a**) Whole specimen´s body on a dark-ground stereomicroscope image. (**b**,**c**) Anterior end; notice oesophagus (es), subtle bulge of the nerve ring (red arrowhead), isthmus of oesophagus (white arrowhead), oesophagus bulb (eb), and intestine (in). Scale bars: (**a**,**b**) 200 µm; (**c**) 50 µm.

**Figure 2 pathogens-10-01152-f002:**
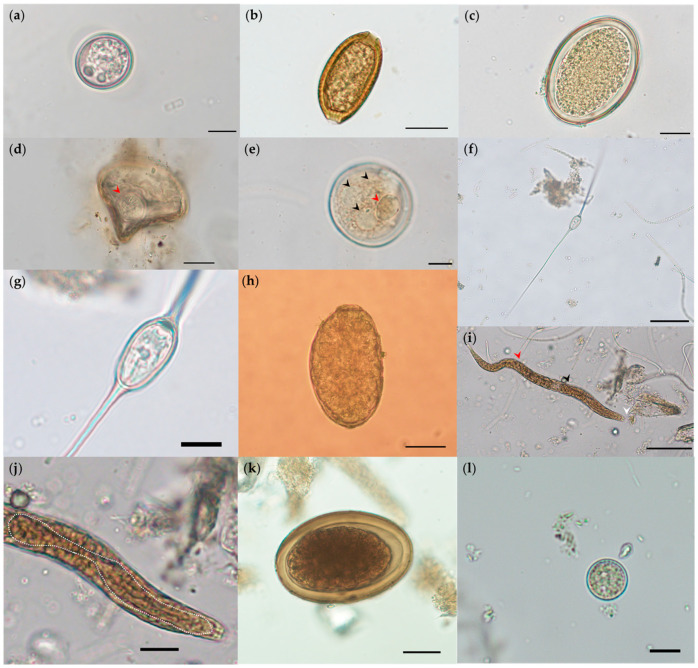
Illustrations of identified parasites stages in fecal samples of free-ranging capybaras (*Hydrochoerus hydrochaeris*) in the Orinoco Basin: (**a**) *Eimeria trinidadensis* oocyst (20 µm × 21.18 µm). (**b**) *Echinocoleus hydrochoeri* (44.72 µm × 24.65 µm) egg. (**c**) *Protozoophaga obesa* egg (70.58 µm × 44.85 µm). (**d**) *Monoecocestus* egg (55.67 µm × 61.84 µm), notice the rounded hexacanth embryo (red arrowhead). (**e**) A *Neobalantidium coli* cyst (46.32 µm × 47.82 µm), notice large macronucleus (red arrowhead) and cytoplasmic contractile vacuoles (black arrowheads). (**f**) *Hippocrepis hippocrepis* egg, notice long filaments (131.92–170.79 µm) and (**g**) miracidium capsule (20.42 µm × 11.27 µm). (**h**) *Taxorchis schistocotyle* egg (157.44 µm × 90.51 µm). (**i**) *Strongyloides*-like larvae (full length 349 µm), notice the rhabditiform oesophagus (87.09 µm; white dotted line), the buccal canal (white arrowhead), oesophageal bulb (black arrowhead) and ventral genital primordium (red arrowhead). (**j**) The strongyloid larvae intestinal esophagus junction width was 19 μm and intestinal undifferentiated cells were noticed. (**k**) Ascarid egg (72.73 µm × 48.89 µm), notice the embryo in advanced cleavage stage. (**l**) *Entamoeba* immature cyst (13.28 µm × 13.44 µm). Scale bars: (**a,g,l**) 10 µm; (**b**–**e,k**) 20 µm; (**f,h,j**) 50 µm; (**i**) 200 µm.

**Figure 3 pathogens-10-01152-f003:**
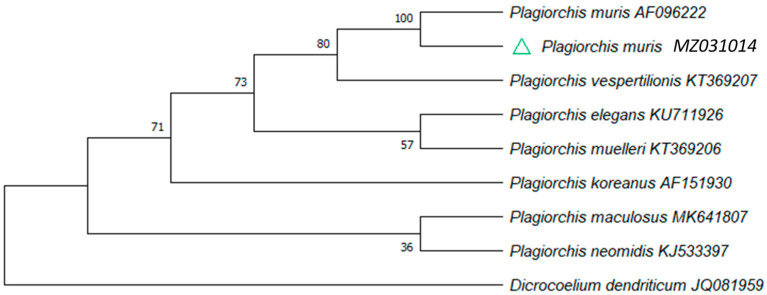
Phylogenetic position of *Plagiorchis muris* isolate obtained from capybara feces. Neighbor-Joining 28S rDNA phylogenic tree. The analysis involved nine nucleotide sequences and all ambiguous positions were removed for each sequence pair (pairwise deletion). The percentage of replicate trees in which the associated taxa clustered together in the bootstrap test (1000 replicates) are shown next to the branches. The specimen collected from capybara feces is indicated by a green triangle.

**Figure 4 pathogens-10-01152-f004:**
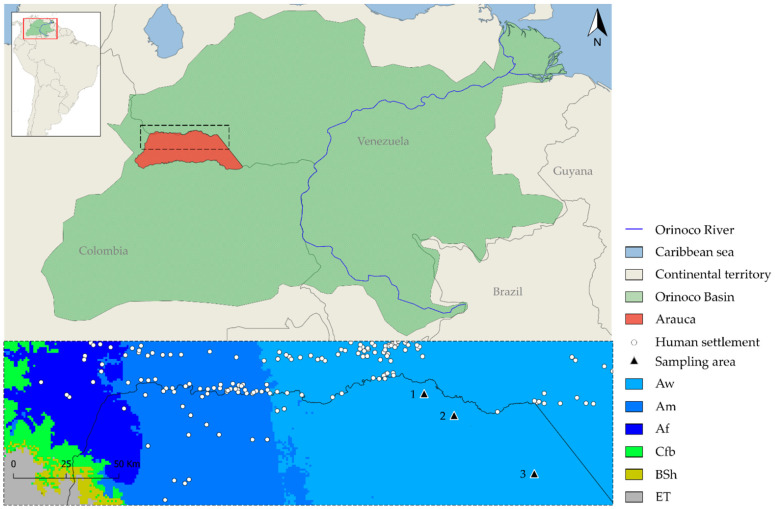
Precise geographic location of sampling areas; Aw: Tropical savanna, Am: Tropical monsoon, Af: Tropical Rainforest, Cfb: Oceanic, BSh: Semiarid, and ET: Tundra. (1) La Maporita, (2) Cinaruco, and (3) Bocas del Arauca.

**Figure 5 pathogens-10-01152-f005:**
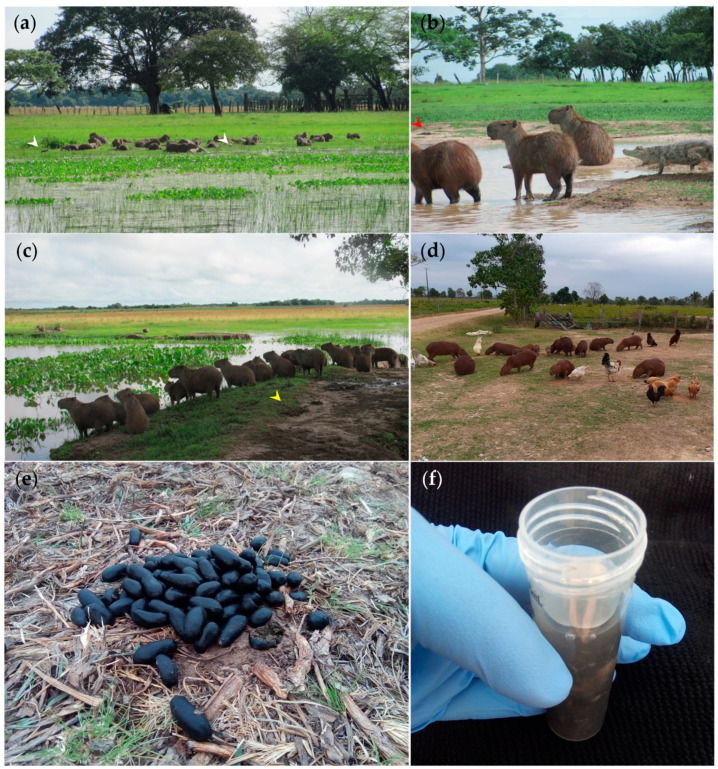
Illustration of free-ranging capybaras (*Hydrochoerus hydrochaeris*) in floodable savannas of the Orinoco Basin. (**a**) Twenty-seven capybara hare resting in Bocas del Arauca mudflats, notice the tight interaction with Charadiiform wader birds (*Vanellus chilensis*) (white arrowheads); (**b**) Small riverside group of capybaras sharing their habitat with spectacled caiman (*Caiman crocodilus*), red arrowhead indicates feces; (**c**) Twenty-three capybara hare in La Maportia, yellow arrowhead indicates feces; (**d**) Peridomestic chickens (*Gallus gallus domesticus*) and synanthropic capybaras feeding in close proximity demonstrating domestic animal–wildlife–human interface; (**e**) Freshly defecated capybara manure pellets piles; and (**f**) EtOH preserved capybara feces.

**Table 1 pathogens-10-01152-t001:** Reported endoparasite species occurring in capybaras (*Hydrochoerus* spp.), including the findings of the present study.

Parasite	Localization ^a^	Tissue	Feces	Blood	Literature
**Metazoa**					
**Nematoda**					
*Dipetalonema hydrochoerus*	C	x			[11]
*Cruorifilaria tuberocauda*	C, Br	x			[12,13,14,15]
*Protozoophaga obesa* ^‡^	A, Br, Bo, C, V		x		[15,16,17,18,19,20,21,22]
*Strongyloides* ^‡^	A, Br, C		x		[16,17,19,22]
*Strongyloides chapini*	Br		x		[15,21]
*Capillaria* spp.	Br		x		[23]
*Echinocoleus hydrochoeri* ^b, ‡^	A, Br, C		x		[15,17,21,22,24]
*Viannella* spp.	Br		x		[19]
*Vianella hydrochoeri*	Br, Bo, V		x		[15,17,18,20,21]
*Hydrochoerisnema anomalobursata*	Br		x		[17,21]
*Trichuris* spp.	Br		x		[17]
*Trichuris cutillasae* n. sp.	A		x		[25]
*Trichostrongylus axei*	Br		x		[15,21]
*Habronema clarki*	Br, Pa		x		[20,26]
*Yatesia hydrochoerus*	Br		x		[15]
Fam: Trichostrongyloidea ^‡^	A, C		x		[22]
Ord: Ascaridida	A		x		[22]
**Cestoda**					
*Monoecocestus* ^‡^	C				Present study
*Monoecocestus hagmanni*	Br, Bo, V		x		[18,20,21]
*Monoecocestus hydrochoeri*	A, Br, Bo		x		[15,16,17,20,21,22]
*Monoecocestus jacobi*	Br		x		[17]
*Monoecocestus macrobursatus*	Br, Bo				[16,20,21]
Fam: Anoplocephalidae	Br		x		[16]
**Trematoda**					
*Fasciola hepatica*	A, Br		x		[23,27,28]
*Hippocrepis fuelleborni*	Br		x		[16]
*Hippocrepis hippocrepis* ^‡^	Br, C, V		x		[15,16,17,18,21,29]
*Hydrochoeristrema cabrali*	Br		x		[17]
*Neocotyle neocotyle*	Br		x		[21]
*Nudacotyle tertius*	Br		x		[15,21]
*Nudacotyle valdevaginatus*	Br		x		[21]
*Plagiorchis muris* ^‡^	C		x		Present study
*Philophthalmus lachrymosus*	Br	x	x		[16,30]
*Taxorchis schistocotyle* ^‡^	A, Br, C, V		x		[15,16,18,21,22]
**Protozoa**					
*Neobalantidium coli* ^‡^	A, C		x		[22]
*Blastocystis* sp.	A		x		[22]
*Cryptosporidium*^‡^/*C. parvum*	B		x		[31]
*Entamoeba* ^‡^	C		x		Present study
*Eimeria* sp./spp.	A, C		x		[22,32]
*Eimeria araside*	Br		x		[33]
*Eimeria boliviensis*	Bo, Br, V		x		[33,34]
*Eimeria ichiloensis*	Bo, Br, V		x		[33,34]
*Eimeria trinidadensis* ^‡^	Bo, Br, C, V		x		[33,34]
*Trypanosoma evansi*	A, Br, Pe, V			x	[35,36,37,38]
*Toxoplasma gondii*	Br			x	[39]
*Giardia* spp.	C		x		[32]
*Sarcocystis* spp.	C		x		[32]
Fam: *Cycloposthiidae* ^‡^	C		x		Present study

^a^ A: Argentina, Bo: Bolivia, Br: Brazil, C: Colombia, Pa: Panama, Pe: Peru, V: Venezuela; ^b^
*Echinocoleus hydrochoeri* syn. *Capillaria hydrochoeri*. ^‡^ Parasites found in this study. x: Parasite location.

**Table 2 pathogens-10-01152-t002:** Occurrence of endoparasites detected in capybaras (*Hydrochoerus hydrochaeris*).

Phylum	Parasite	Stage ^a^	Technique ^b^	Bocas del Arauca *n* = 15	Cinaruco *n* = 8	La Maporita *n =* 23	Total (%)
Apicomplexa	*Cryptosporidium*	O	coproELISA	2	6	8	34.8 (16/46)
	*Eimeria trinidadensis*	O	SAF	2	2	6	21.7 (10/46)
Amoebozoa	*Entamoeba*	C	SAF	4	2	2	17.4 (8/46)
Ciliophora	*Neobalantidium coli*		SAF	1		1	4.3 (2/46)
	Cycloposthiidae	C	SAF	6		4	21.7 (10/46)
Platyhelminthes							
Nematoda	Ascarididae	E	SAF	3	2	8	28.3 (13/46)
	*Echinocoleus hydrochoeri*	E	CF/SAF	7	7	13	58.7 (27/46)
	*Protozoophaga obesa*	E/L/A	SS/CF/SAF	3	1	4	17.4 (8/46)
	*Strongyloides*-like	L	SAF	7	2	10	41.3 (19/46)
Cestoda	*Monoecocestus*	E	CF/SAF	1		2	6.5 (3/46)
Trematoda	*Hippocrepis hippocrepis*	E/A	SF/SS/SAF	4	1	3	17.4 (8/46)
	*Plagiorchis muris*	A	Sequencing	1			2.2 (1/46)
	*Taxorchis schistocotyle*	E	SS/SAF	6	6	8	43.5 (20/46)

^a^ O: oocysts, C: cysts, E: eggs, L: larvae, A: adult; ^b^ SF: sedimentation–flotation, SAF: modified sodium acetate–acetic acid–formalin, SS: simple sedimentation, CF: centrifugal flotation, CFS: fast carbol–fuchsin stained fecal smear.

## Data Availability

The data presented in this study are available in the Appendix A. The putative *Plagiorchis muris* sequence obtained from capybara (*Hydrochoerus hydrochaeris*) feces was deposited in the GenBank database (National Center for Biotechnology Information, NIH, Bethesda, USA), and are simultaneously available at ENA (European Nucleotide Archive) in Europe and the DDBJ (DNA Data Bank of Japan) under accession number MZ031014.

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
