# Peer review of "Parasites Circulating in Wild Synanthropic Capybaras (Hydrochoerus hydrochaeris): A One Health Approach"

_pathogens, 2021, doi:10.3390/pathogens10091152_

Round 1
Reviewer 1 Report
This study examines endoparasites and their zoonotic potentials in capybaras, a good example of One Health study. The paper is generally clear and well written except that the descriptions on each parasite in Results are unnecessarily lengthy.
P73-74: “Currently, multiple endogenous proto- and metazoan parasites have been described in capybaras”, please list the related references for parasite study on capybaras
P87-88: “comprising 5 protozoans of the phyla Apicomplexa, Amoebozoa and Ciliophora”, but in Table 1, only four protozoan species were marked with ‡.
P197: change Plagiorchid to Plagiorchis
P214-215: “Synanthropic capybaras play a preservation key role in fragile neotropical ecosystems”, changing to play a key role in preserving fragile neotropical ecosystems
P217-219: “It is estimated that approximately 60% of emerging human infections are zoonotic and among these pathogens more than 70% originated from wildlife species”. Please instead cite the original article “Taylor, L.H., Latham, S.M. and Woolhouse, M.E., 2001. Risk factors for human disease emergence. Philosophical Transactions of the Royal Society of London. Series B: Biological Sciences, 356(1411), pp.983-989”, which instead stated that 61% of human diseases are zoonotic, and out of the “emerging” pathogens, 75% are zoonotic. Rahman et al. cited by this article also states that more than 60% of human pathogens (but not emerging human pathogens) are zoonotic in origin
P253-254: “Cinaruco 75% (6/8), furthest away from populated centers whose closest human settlement is located at 33738 Km”, do you mean the human population is 33738 Km away? I think this is impossible based on the lower panel of figure 5.
P31, 290, 292, 296, 299: change lagochilascariosis to lagochilascariasis
P341: “90% of environmental humidity”, unclear the meaning of environmental humidity, do you mean relative humidity?
P343-344: “Individual faecal samples were collected during dry season from March to June 2020 in La Maporita”, why not collecting samples during the rainy seasons when the gastrointestinal parasite infection rate may be higher?
Author Response
Dear Reviewer;
We would like to thank you for highlighting the One Health focus of the research approach presented here and the manuscript comments, all of them in favor to improve it. Regarding the descriptions of parasite stages morphology, we seriously consider abbreviating them and just present the more relevant taxonomic traits as you recommended. Along the text when we cite “Currently, multiple endogenous proto- and metazoan parasites have been described in capybaras” (Lines 73-74), now we appropriately refer the reader to Table 1 were a complete list of parasitological studies on capybaras is given. The corresponding number of reported protozoan parasite of the phyla Apicomplexa, Amoebozoa and Ciliophora cited in Lines 87 and 88 were correctly marked (‡) in Table 1. The Line 216 was changed to “play a key role in preserving fragile neotropical ecosystems” as equal as the Lines 220-223 were we made the amendments in the text and the proper citation of Taylor et al 2001 as such you indicate us.
Additionally, through a GIS software we recheck the distance between Cinaruco sampling area and the nearest human settlement. You were completely right; we have indeed made a syntax and accuracy mistake in the distance. The new version of the Figure 5 second panel now includes an adequate scale bar and the accurate distance in Km was rectified in the text (i. e.; 20.598Km).
Finally, the relative humidity concept was clarified in line 345, “Plagiorchid” was changed to “Digenean” in Line 199 and the term “lagochilascariosis” was changed along the text to “lagochilascariasis”. Concerning the last specific point, it is important to take into account that during rainy season the floodable savannas and lowlands of the Orinoco Basin are inaccessible terrain by car and thus the sampling of those free-ranging rodent will be even more challenging and highly risky due to flooded areas.
PS: Attaches below you will find a short cover letter which detailed the changes along the text.
Best regards.

Reviewer 2 Report
In this manuscript the authors report the results of their parasitological examination of 46 fecal samples from Capybaras and discuss their results in the context of earlier literature, with special emphasis on putative zoonotic transmission of the parasites from Capybaras to humans. The manuscript is generally well written and understandable. The authors found 15 different protozoan and helminth parasites and were able to identify them to various taxonomic levels. Five of these parasites had not been reported from Capybaras before, among themPlagiorchis muris, a known zoonotic parasite, for which this is the first report in South America. For this species the authors used molecular markers to assist species determination. The value of the study would be significantly increased, if the same had been done for other species found. Some of the implications for zoonotic potential the authors make, in particular in the title and the abstract, appear a bit exaggerated given that many of the parasites could only be identified to a higher taxonomic level. Molecular information could add significantly to this discussion, in particular for taxa for which corresponding information from human derived parasites is available. Nevertheless, the data presented are interesting and appear technically sound. However, due to the rather small size of the survey (overall and in particular for each of the three locations) and the absence of molecular data (except for P. muris), the amount of novel information is limited. These data deserve to be on the record but, in my opinion, publishing them neither requires nor justifies such a long paper with 19 printed pages and more than 90 references in a general journal. I suggest that the authors publish these results along with only the essential introduction and discussion in a much more concise form. If "Pathogens" is the right place to do so, is up to the editors to decide, in my opinion, a specialized parasitological journal would be more appropriate. The authors did compile a large body of literature and discuss it in this manuscript. This is in principle interesting and useful. However, this manuscript, as it stands, is neither a concise report of a parasitological survey nor a true review article. The authors might consider writing, in addition, a review article, in which they summarize the relevant current knowledge (and lack thereof) about endoparasites in capybaras and discuss the putative role of these animals for zoonotic transmission of pathogens. Such a review would include, but not need, to focus on the authors own data and it would not be burdened by experimental details. I could imagine that such a review article would fall within the interest of "Pathogens".
Specific points:
1) Lines 26-28: this claim is exaggerated. Some of these parasites were only identified to a higher taxonomic level and it can therefore not be evaluated if they are likely zoonotic or not. For example, the vast majority of Strongyloidesspp., among them all species described in rodents I am aware of, are not zoonotic. Molecular data would be most helpful.
2) Lines 32,33: judging from Table 1, Monoecocestussp. and Fam Taeniidae had also not been reported from Capybaras before.
3) Line 46-50: This sentence is grammatically incorrect (it has no verb).
4) Table 1 / Table 2: Cyclophosthium sp. is included in Table 2 but missing in Table 1
5) Table 2: Given the small sample sizes, I do not think it is justified to give % for occurrence. Absolute numbers would be more appropriate.
Author Response
Dear Reviewer;
First, we would like to thank you for highlighting the recent cited refences, the human-animal interface as the main focus of current research approach, and the usefulness of presented information. The text was subjected to an extensive review, syntax double-check, illative use of English, and accurate reference quoting. Misleading phrases were rewording and edited to clearly develop the main idea. Regarding the extension of the manuscript, we made it more concise. A thoroughly review of the morphological description of parasite stages has been made, thus presenting the more characteristic taxonomical traits avoiding over description of well-known parasites. The title was changed to a more suitable and concordant version based on your comments. The correspond grammatically amendment in lines 46-50 was made and the Cyclophosthium species was included in Table 1 as such you indicate us. You were absolutely right, any Taeniid had not been reported in those giant rodents but M. hagmanni, M. hydrochoerid, M. jacobi, and M. macrobursatus have been previously reported. Based on your suggestion, Table 2 present the absolute numbers of parasite in sampling areas, thus using % only for total occurrence.
Even a small sample size in free-ranging neotropical wildlife constitute a huge sampling effort considering the inhospitable terrain and animals´ elusive nature, but we hope that this kind of approach will be welcome in the special issue "Parasitic Diseases of Domestic, Wild, and Exotic Animals” which belongs to the section "Parasitic Pathogens". We will be delighted to contribute with any kind of further invited review in wildlife parasites or NTD parasitic agents. Overall, the sixteen parasite taxa here identified, eight parasite stags were described to species level, six to genus and finally just two of them to a higher taxonomic level (i. e., family). Regarding the zoonotic potential of parasitic NTD circulating in free-ranging capybara populations further studies are explicitly suggested for a better understanding, not only of their neotropical lifecycles, but also of their eco-epidemiology and public health concern issue. The possible presence of animal-associated Strongyloides spp. such as the “nutria itch” should not be ruled out in capybara identified nematodes, although they are not considered “true” strongyloidiasis in humans.
PS: Attaches below you will find a short cover letter which detailed the changes along the text.
Warmest regards.

Round 2
Reviewer 2 Report
After seeing the changes the authors made and the comments by the other reviewer and the editor, I support publication of the manuscript, essentially in its current form. While reading, I noticed a few very minor things that can be corrected during the production process. They are listed below.
Line 27 first word: "parasites" not "parasities"
Line 228: "as" missing after "such"
Lines 245,246: This sentence lacks some essential words. I propose: "Furthermore, here we identify P. muris. Within the Plagiorchiidae family, this is the only species capable of infecting humans and it has been reported across continents [56]."
Line 261: "space" is missing before "4.3" and "%" is missing after "4.3"
Line 263: "felids" (plural)
Lines 291-295: This paragraph is not really clear. Why is strongyloidiasis neglected due to its auto -infective cycle? Isn't it rather due to the fact that it is frequently not diagnosed and its prevalence is therefore under estimated??
Author Response
Dear Reviewer;
First, we would like to thank you for such a precise and clear text corrections all of them in favor to improve the manuscript quality. The word “parasities” has been changed to “parasite” in line 27. The sentence in lines 272-276 was replaced to “… Within the Plagiorchiidae family, this is the only species capable of infecting humans and it has been reported across continents…” as you propose. The amendment of missing "space" and absence of “%” in line 292 was made as equal as the plural form of the word in line 294. Finally, regarding the strongyloidiasis phrase in lines 324-325 has been reworded too.
The new grammatically amended manuscript version could be found attached below.
With kind regards.